# Prognostic Factors of Survival in Glioblastoma Multiforme Patients—A Retrospective Study

**DOI:** 10.3390/diagnostics12112630

**Published:** 2022-10-30

**Authors:** Georgiana Șerban, Flaviu Tămaș, Rodica Bălașa, Doina Manu, Corina Tămaș, Adrian Bălașa

**Affiliations:** 1Doctoral School, “George Emil Palade” University of Medicine, Pharmacy, Science and Technology of Targu Mures, 540142 Targu Mures, Romania; 2Neurosurgery Clinic, Emergency Clinical County Hospital of Targu Mures, 540136 Targu Mures, Romania; 3Department of Neurosurgery, “George Emil Palade” University of Medicine, Pharmacy, Science and Technology of Targu Mures, 540136 Targu Mures, Romania; 41st Neurology Clinic, Emergency Clinical County Hospital of Targu Mures, 540136 Targu Mures, Romania; 5Department of Neurology, “George Emil Palade” University of Medicine, Pharmacy, Science and Technology of Targu Mures, 540136 Targu Mures, Romania; 6Center for Advanced Medical and Pharmaceutical Research, “George Emil Palade” University of Medicine, Pharmacy, Science and Technology of Targu Mures, 540136 Targu Mures, Romania

**Keywords:** glioblastoma multiforme, survival rate, prognostic markers

## Abstract

Background: Glioblastoma multiforme (GBM) is the most aggressive brain tumor that occurs in adults. In spite of prompt diagnosis and rapidly administered treatment, the survival expectancy is tremendously poor. Extensive research has been performed in order to establish factors to predict the outcome of GBM patients; however, worldwide accepted prognostic markers are still lacking. Methods: We retrospectively assessed all adult patients who were diagnosed with primary GBM and underwent surgical treatment during a three-year period (January 2017–December 2019) in the Neurosurgery Department of the Emergency Clinical County Hospital of Târgu Mureș, Romania. Our aim was to find any statistically relevant connections between clinical, imagistic, and histopathological characteristics and patients’ survival. Results: A total of 75 patients were eventually included in our statistical analysis: 40 males and 35 females, with a median age of 61 years. The mean tumor dimension was 45.28 ± 15.52 mm, while the mean survival rate was 4 ± 6.75 months. A univariate analysis demonstrated a statistically significant impact of tumor size, pre-, and postoperative KPSI on survival rate. In addition, a Cox multivariate assessment strengthened previous findings regarding postoperative KPSI (regression coefficient −0.03, HR 0.97, 95% CI (HR) 0.96–0.99, *p* = 0.002) as a favorable prognostic factor and GBM size (regression coefficient 0.03, HR 1.03, 95% CI (HR) 1.01–1.05, *p =* 0.005) as a poor prognostic marker for patients’ survival. Conclusions: The results of our retrospective study are consistent with prior scientific results that provide evidence supporting the importance of clinical (quantified by KPSI) and imagistic (particularly tumor dimensions) features as reliable prognostic factors in GBM patients’ survival.

## 1. Introduction

Glioblastoma multiforme (GBM), a WHO grade IV central nervous system tumor, is one of the most aggressive malignancies occurring in adults [1]. Currently, the standard treatment, also known as the Stupp protocol, consists of extensive surgical removal, adjuvant radiotherapy, and chemotherapy with temozolomide [2]. Nonetheless, GBM possesses remarkable, although incompletely recognized, ways to elude the therapeutic methods. Due to its highly invasive nature, complete surgical removal at the cellular level is practically unachievable and recurrences are inevitable [1,2]. Furthermore, although the blood-brain barrier in GBM patients has an abnormally higher permeability, its disrupted structure is not uniformly distributed and, as a result, the chemotherapeutic molecules hardly attain the desired therapeutic level within tumor cells [1]. Moreover, GBM patients have a meagre immune response by promoting a tumor-induced anti-inflammatory response and creating an immunosuppressive environment for tumor growth [1,2]. Consequently, regardless of how rapidly the diagnosis is determined and how the treatment is initiated, the overall survival is extremely low, i.e., only 12–18 months [3]. 

Regarding GBM prognosis, several markers have been taken into consideration. According to the WHO classification, there are two subtypes of GBM depending on their respective genetic features, particularly, the presence or absence of IDH mutations: primary and secondary GBMs [3,4]. On the one hand, primary GBMs usually occur in elderly patients, have no precursor benign or malignant lesions and generally yield a worse prognosis [2,4,5]. On the other hand, secondary GBMs appear at a younger age, typically develop from lower grade gliomas, are associated with IDH mutations, and have a longer overall survival [3]. Furthermore, it has been demonstrated that different histopathological subtypes respond differently to treatment and, consequently, have distinct survival rates [6]. Additional prognostic markers include patient age at disease onset, clinical markers (e.g., preoperative Karnofsky performance index), imagistic aspects (e.g., dimension and location of the tumor, size of necrosis, and edema surrounding the tumor), and the extent of surgical removal [6,7]. 

On these grounds, our retrospective study aimed to assess the influence of clinical, imagistic, and histopathological markers on the survival of patients diagnosed with primary glioblastoma.

## 2. Materials and Methods

We performed a retrospective study during a three-year period from January 2017 to December 2019. Prior to data collection and analysis, ethical consent was obtained from the Local Ethics Committee of the Emergency Clinical County Hospital of Târgu Mureș, Romania (no. 21490/09.09.2022). Our study comprised 89 patients who had a confirmed histopathological diagnosis of GBM and who were surgically treated during the abovementioned period in the Neurosurgery Department of the Emergency Clinical County Hospital of Târgu Mureș, Romania. The inclusion criteria consisted of adult patients at the moment of the surgery, confirmed histopathological diagnosis of primary GBM, and surgical removal of the tumor. The patients, who were diagnosed with secondary GBM, refused surgical intervention, were suitable only for biopsy due to high surgical risk, had previous history of malignancies or radio- and chemotherapy, or whose clinical and imagistic data could not be found, were excluded from further statistical analysis. All patients received adjuvant postoperative radiotherapy and chemotherapy with temozolomide. 

Using the patients’ medical records from the Hospital Information System and the verbal and written information delivered from the patients’ relatives, we obtained the following data: gender and age of the patients at the moment of diagnosis; dimension, location, molecular subtype, and immunohistochemical characteristics of the tumor; pre- and postoperative Karnofsky Performance Scale Index (KPSI); extent of surgical ablation; overall survival (hereby, defined as the time in months from the first surgical intervention to the time of death). The tumor size refers to the largest extension in axial direction based on most recent preoperative MRI images and is expressed in millimetres (mm). Based on postoperative control scans, the extent of resection was categorized into three groups: biopsy, subtotal ablation (<90% of the tumor removed), and total ablation (≥90% of the tumor removed). Regarding immunohistochemical data, we evaluated the presence of the Ki67 index and glial fibrillary acidic protein (GFAP) expression. 

The data were summarized in a Microsoft Excel spreadsheet and further statistically assessed using the GraphPad and MedCalc software programs. We used the Kolmogorov–Smirnov normality test in order to determine the distribution for numerical data (e.g., age, Karnofsky Performance Scale Index, tumor size). In the case of Gaussian distribution, we further presented the data as means ± standard deviation, and according to the data type (paired or unpaired), applied the variants of a Student’s *t*-test for two samples or the variants of an ANOVA test for at least three samples, respectively. In the case of non-parametrical distribution, we expressed the data as medians ± interquartile range, and depending on the data type (paired or unpaired), applied the Wilcoxon and Mann–Whitney tests for two samples or the Friedman and Kruskal–Wallis tests for at least three samples, respectively. When comparing means or medians for at least three samples and obtaining statistically significant results, we further applied post hoc tests in order to provide more reliable results by controlling the type 1 error rate: Tukey post hoc test for data with Gaussian distribution and Dunn post hoc test for data with non-Gaussian distribution. The Fisher exact test was chosen to compare independent and dependent variables, and therefore, find potential associations, notably because it is far more precise than chi-square test when the values within the tables are small. In order to assess the potential correlation between two samples, we calculated either Pearson’s, or Spearman’s correlation coefficients, in accordance with data distribution. Overall survival rates were calculated using Kaplan–Meier survival curves. We compared different survival curves using a log-rank test. We utilized a Cox regression test to assess the influence of independent variables on overall survival. We established a threshold of statistical significance of 0.05, thus, a *p*-value lower than 0.05 was considered to be statistically significant, with a confidence interval of 95%.

## 3. Results

From January 2017 to December 2019, 89 patients with histopathologically confirmed diagnoses of GBM were hospitalized in the Neurosurgery Clinic of The Emergency Clinical County Hospital of Târgu Mureș. Fourteen patients were excluded from further statistical analysis due to exclusion criteria, including five patients that only underwent biopsies, two patients diagnosed with secondary GBM, and seven patients with no found follow-up data. Among the remaining 75 cases, there were 40 (53.33%) males and 35 (46.67%) females, with a median age of 61 years. The mean tumor size was 45.28 ± 15.52 mm. Most commonly, GBM developed in either the frontal or temporal area of the brain, whereas the occipital region was rarely affected—only one case out of 75. There was an approximately equal distribution of the tumors between left and right hemisheres (40 and 35 out of 75 tumors, respectively). Since all the patients were right handed, the left hemisphere was considered to be the dominant one. The pre- and postoperative Karnofsky Performance Scale Index (KPSI) medians were both 80. Regarding the extent of tumor removal, total ablation was performed radiographically in 92% of the cases. The patients had a median overall survival of 4 ± 6.75 months. Only three patients were still alive at the moment of data collection (September 2022): one patient with epitheloid GBM and two patients with NOS GBM. The most frequent molecular subtype was NOS GBM (38 out of 75 patients), while the least common were IDH mutant GBM and gliosarcoma, with three patients each. Twelve cases belonged to IDH wild type GBM and 19 cases belonged to epitheloid GBM. No statistically significant differences regarding age, tumor dimension, pre- and postoperative KPSI, and survival were recorded among various molecular GBM subtypes (Table 1). As far as the immunohistopathological analysis was concerned, 29 (38.67%) patients had an increased level of Ki67 index (≥25%). Nonetheless, we could not assess Ki67 in almost half of the cases due to the unavailability of data (34 out of 75 cases). In our study group, 49 (65.33%) patients had positive GFAP staining. 

Furthermore, we tried to find any associations between the involvement of a certain cerebral hemisphere and the extent of surgical ablation, yet no statistical significant result was obtained (Table 2).

We also assessed the statistical correlations among age, tumor size, and the pre- and postoperative KPSI of the patients with similar molecular GBM subtypes and of the total GBM population. Our results showed a strong positive correlation between pre- and postoperative KPSI in both IDH wild type (Spearman r *=* 0.76, *p =* 0.001) and NOS GBM (Spearman r 0.608, *p* < 0.0001) groups. In addition, we demonstrated a medium negative correlation between age and preoperative KPSI (Spearman r *=* −0.457, *p =* 0.003) and age and postoperative KPSI (Spearman r *=* −0.402, *p =* 0.012), in the NOS GBM subpopulation. No other statistically significant correlations were found (Table 3).

Using the Kaplan–Meier survival curves and log-rank tests, we compared the estimates of clinical outcomes among different molecular subtypes (Figure 1a), tumor location (Figure 1b), involvement of dominant versus non-dominant cerebral hemisphere (Figure 1c), extent of ablation (Figure 1d), GFAP staining (Figure 1e), and Ki67 index level (Figure 1f). Nonetheless, no significant differences were found among the groups in terms of survival. The univariate analysis performed on all GBM cases revealed a significant impact on survival for tumor dimension (*p* < 0.001), preoperative KPSI (*p* = 0.003), and postoperative KPSI (*p* < 0.001). Age, gender, molecular subtype, GBM location, involvement of either dominant or non-dominant cerebral hemisphere, extent of ablation, and immunohistochemical staining did not yield any influence on patients’ survival. The Cox multivariate analysis demonstrated postoperative KPSI (regression coefficient −0.03, HR 0.97, 95% CI (HR) 0.96–0.99, *p* 0.002) as a favorable prognostic factor, while GBM size (regression coefficient 0.03, HR 1.03, 95% CI (HR) 1.01–1.05, *p* 0.005) was considered to be a poor prognostic marker for patients’ survival. Other variables taken into consideration showed no statistical significance. When each molecular subtype was assessed individually, the univariate analysis showed that tumor dimension influenced survival of the patients diagnosed with IDH wild type and NOS GBM. Moreover, in both epitheloid and NOS GBM, postoperative KPSI had a significant impact on survival. Nonetheless, the Cox multivariate regression shared similar results with those obtained in the whole GBM population, as in, the IDH wild type group tumor size proved to be a negative prognostic factor (regression coefficient 0.13, HR 1.14, 95% CI (HR) 1.01–1.13, *p* = 0.034), whereas, in epitheloid subpopulation, postoperative KPSI appeared to be a favorable prognostic marker (regression coefficient −0.1, HR 0.9, 95% CI (HR) 0.83–0.99, *p* = 0.025) for patients’ survival. 

## 4. Discussion

GBM is one of the most aggressive brain disorders and represents a major threat to patients’ quality of life and, eventually, to their survival [8]. Research that focuses on molecular markers intended to estimate GBM prognosis is still in its infancy [9]. Until these markers demonstrate their clinical efficacy and accuracy and become available on a large scale, clinical, imagistic, and histopathological indicators could be utilized in order to predict the evolution of this disease in each individual [10]. Our study results demonstrated, with strong statistical significance, that tumor size and the Karnofsky Performance Scale Index (KPSI) calculated both pre- and postoperatively were linked to patients’ survival. Furthermore, in the NOS molecular subtype, age was directly correlated to the Karnofsky score. 

An increasing trend has been demonstrated in terms of onset age due to both a higher incidence of GBM in elderly patients and prolonged lifespan of people [11]. The current median age is around 64 years [11,12]. In our study, the mean age (61 years) was comparable with that stated in the literature. Voisin et al. [12] developed a study in which the purpose was to establish the clinical predictors of survival in elderly GBM patients. Their results showed that postoperative complications were associated with increased age and a lower survival. Moreover, Marton et al. [13] confirmed that a younger age at the moment of diagnosis was related to a more favorable outcome for the patients. Similarly, our study demonstrated that age was negatively correlated with postoperative KPSI, a scoring system which mirrors the clinical status of the patients. Nonetheless, our results also showed a negative correlation between age and preoperative KPSI, although this was limited to the NOS subgroup. 

Most GBMs occur in the supratentorial compartment of the brain, particularly, in the frontal, temporal, and parietal lobes. Tumors in the occipital lobe, posterior fossa, and spinal cord are uncommon [14]. Simpson et al. [15] demonstrated a higher survival for patients with frontal lobe GBM, most likely due to molecular features (namely, IDH mutant subtypes) and a more extended surgical removal [16,17,18,19]. Our study revealed a prevalent appearance in the frontal and temporal lobes, yet no significant difference in patients’ survival was reported. Furthermore, we noted similar survival rates among various molecular subtypes. However, the distribution of the patients in different molecular classes was fairly uneven, thus, the statistical value might be questionable. The extent of ablation also did not reveal any disparity in patients’ survival, although this result should be cautiously regarded as the great majority of our patients had a total removal of the tumor. There are numerous studies that have reported the benefit of a more extensive tumor removal on a longer survival for GBM patients, particularly, by providing a more suitable environment for the efficacy of adjuvant treatment [20,21,22]. 

Immunohistochemical staining is routinely assessed when performing a histopathological diagnosis of GBM. In our statistical analysis, we used glial fibrillary acidic protein (GFAP) and the Ki67 index. Ahmadipour et al. [23] established a cut-off value of 75% for GFAP and predicted that patients with GFAP ≥ 75% had lower long-term survival. Moreover, Gallego Perez-Larraya et al. [24] demonstrated a significant correlation between GFAP expression in GBM and tumor size. We qualitatively assessed GFAP and discovered no difference in survival between patients with and without GFAP expression. Moreover, although we did find a positive correlation between GFAP expression and tumor dimension, it was not statistically significant. The Ki67 index, a marker which quantifies proliferation [14], has been taken into consideration as a predictor marker for poor survival in malignant tumors. Alkhaibary et al. [25] used a cut-off value of 27%, but they did not prove a significant difference between survival curves of the patients with or without elevated levels of Ki67 index. We also did not manage to link a high Ki67 index to a poorer survival while using a cut-off value of 25%. Armocida et al. [26] demonstrated that a Ki67 proliferation index over 20% forecasted a poorer progression free survival in IDH wild type molecular subtype. To add more uncertainty on this matter of debate, Wong et al. [27] achieved a positive correlation between the Ki67 index and overall survival; a Ki67 index less than 22% predicted a more reduced overall survival in GBM patients. 

Multiple studies have already demonstrated that large preoperative tumor dimensions have an important negative impact on patients` survival [6,10,28,29,30]. Inoperable GBMs have a reduced survival expectancy of only weeks to a few months; this period of time could be increased to maximum 10 months by solely surgical removal of the tumor [31]. The Stupp protocol (surgery followed by adjuvant chemo- and radiotherapy) could improve survival to a total of at least 14 months on average [32]. The mean survival in our study group was 8 months, although the survival range was extremely large, from less than 2 weeks postoperatively to patients that were still alive even after 5 years from surgery. Taking into consideration that all our patients followed adjuvant therapy in addition to surgical intervention, the survival rate was only half of what we had expected. Since demographic, clinical, and histopathological features of our patients were comparable with those in the scientific reports, we hypothesized that tumor dimension might play a paramount role in life expectancy of our GBM patients. The univariate and multivariate analyses we performed confirmed that GBM size was a statistically significant negative prognostic for patients’ survival. Whitmire et al. [33] conducted a study which assessed gender-specific influences of various clinical, demographical, and imagistic factors on survival in GBM patients. They defined extreme survivors (overall survival of 5 years or more) and short-term survivors (overall survival of less de 210 days). Among others, they concluded that tumor dimensions influenced the overall survival in GBM patients. Nonetheless, both categories, regardless of gender, had mean tumor dimensions smaller than those described in our study, which might, at least partially, explain the reduced overall survival in our study population. Another possible explanation might relate to the involvement of the dominant cerebral hemisphere which could worsen preoperative and postoperative KPSI and limit the degree of surgical removal. However, our study showed no statistically significant difference in terms of survival when assessing the involvement of either cerebral hemisphere. On the one hand, Polin et al. [34] found that tumor lateralization did not influence either the functional outcome of GBM patients or the overall survival. On the other hand, Coluccia et al. [35] showed a shorter progression-free period and a bigger reduction in postoperative KPSI for left hemisphere GBM, most likely due to a more cautious surgical approach. Further research should be conducted in order to draw a reliable conclusion, and thus, offer the most suitable surgical treatment for GBM patients.

## 5. Limitations

One of the main acknowledged limitations of our study is the single center implementation and, consequently, the reduced number of patients involved. A multicenter study would have allowed a wider selection of patients with a higher cohort for each molecular subtype and a much more even distribution within various subgroups for further statistical analysis. Moreover, although we did know that our patients had undergone adjuvant therapy in addition to surgical removal, supplementary assessment of duration and doses of chemo- and radiotherapy would have permitted a broader perspective on disease evolution. Another component that would have increased the scientific value of our study would be a more detailed immunohistochemical analysis of the brain samples. Furthermore, the retrospective design of our study impeded a closer follow-up of the patients. 

## 6. Conclusions

Glioblastoma is the most aggressive brain tumor that occurs in adults, with a dreadfully low life expectancy. Currently, there are no reliable prognostic markers, although molecular studies that aim to find such markers are being performed. At this point, only clinical, histopathological, and imagistic factors can predict the survival of GBM patients. Our study strengthens the results of previous publications that have reported the influence of tumor dimension on survival rate. We further demonstrate a significant negative correlation between age and pre- and postoperative clinical status of the patients, quantified using the Karnofsky Performance Scale Index (KPSI), as well as a noteworthy pre- and postoperative KPSI impact on survival rate. 

## Figures and Tables

**Figure 1 diagnostics-12-02630-f001:**
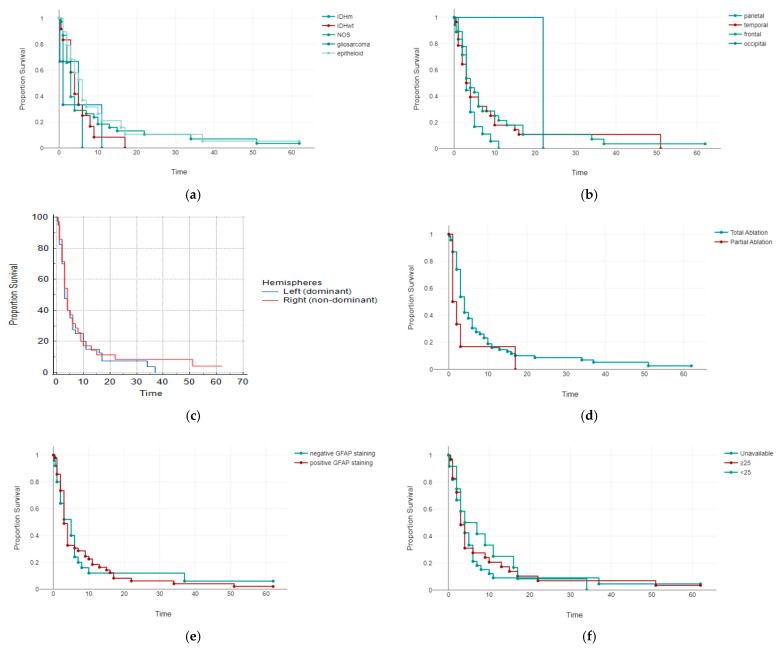
Comparison of Kaplan–Meier estimates of clinical outcomes between the patients with: (**a**) Different molecular subtypes (log-rank *p* = 0.665); (**b**) different locations of the tumors (log-rank *p* = 0.241); (**c**) involvement of dominant versus non-dominant cerebral hemisphere (log-rank *p* = 0.604); (**d**) total and partial tumor ablation (log-rank *p* 0.16); (**e**) with positive and negative GFAP staining (log-rank *p* = 0.968); (**f**) different levels of Ki67 index (log-rank *p* = 0.792).

**Table 1 diagnostics-12-02630-t001:** Descriptive statistics (data with Gaussian distribution are presented as arithmetic means ± standard deviation and data with non-Gaussian distribution are presented as medians (Q1; Q3)); * Too few values to calculate normality, data are presented as arithmetic means ± standard deviation).

Molecular Subtype	Total Number	Gender	Age (Years)	Tumor Dimension (mm)	Preoperative KPSI	Postoperative KPSI	Survival (Months)
* IDH mutant	3 (4%)	M: 3 (100%)	59 ± 7.55	62 ± 19.52	86.67 ± 5.77	80 ± 20	4 ± 2.65
		F: 0 (0%)					
IDH wild type	12 (16%)	M: 5 (41.67%)	57.91 ± 16.56	41.16 ± 14.32	84.16 ± 9	90 (80; 95)	5.29 ± 4.46
		F: 7 (58.33%)					
NOS	38 (50.67%)	M: 21 (55.27%)	63.5 (50; 69)	45.77 ± 16.45	80 (60; 90)	80 (70; 90)	3 (2; 9)
		F: 17 (44.73%)					
* Gliosarcoma	3 (4%)	M: 2 (66.67%)	42 ± 7.93	57 ± 15.39	40 ± 34.64	50 ± 30	4 ± 6
		F: 1 (33.33%)					
Epitheloid	19 (25.33%)	M: 9 (47.36%)	57.73 ± 12.23	42.41 ± 12.28	80 (60; 80)	80 (80; 90)	6 (3; 10.75)
		F: 10 (52.64%)					
All GBM subtypes	75 (100%)	M: 40 (53.33%)	61 (48.5; 68.75)	45.28 ± 15.52	80 (70; 90)	80 (80; 90)	4 (2; 8.75)
		F: 35 (46.67%)					

**Table 2 diagnostics-12-02630-t002:** Fisher exact test assessing for a possible association between involvement of a certain cerebral hemisphere and the extent of surgical removal of the tumor (*p* 0.678, OR 0.545, 95% CI 0.093–3.178).

	Total Ablation	Partial Ablation
Left Hemisphere	36 (48%)	4 (5%)
Right Hemisphere	33 (44%)	2 (3%)

**Table 3 diagnostics-12-02630-t003:** Statistical correlations among different variables in GBM patients.

Molecular Subtype	Age—Tumor Size	Age—Preoperative KPSI	Age—Postoperative KPSI	Tumor Size- Preoperative KPSI	Tumor Size—Postoperative KPSI	Preoperative KPSI—Postoperative KPSI
IDH mutant	*p* = 0.36	*p* = 0.36	*p* = 0.36	*p* = 0.36	*p* = 0.36	*p* = 0.36
	Pearson r 0.84	Pearson r 0.84	Pearson r 0.84	Pearson r 0.84	Pearson r 0.84	Pearson r 0.84
IDH wild type	*p* = 0.55	*p* = 0.23	*p* = 0.79	*p* = 0.55	*p* = 0.54	*p* = 0.0016
	Pearson r −0.17	Pearson r −0.34	Spearman r −0.074	Pearson r −0.17	Spearman r 0.18	Spearman r 0.7615
NOS	*p* = 0.34	*p* = 0.003	*p* = 0.012	*p* = 0.09	*p* = 0.13	*p* < 0.0001
	Spearman r 0.157	Spearman r −0.4576	Spearman r −0.4029	Spearman r −0.277	Spearman r −0.247	Spearman r 0.608
Gliosarcoma	*p* = 0.18	*p* = 0.18	*p* = 0.18	*p* = 0.18	*p* = 0.18	*p* = 0.18
	Pearson r −0.956	Pearson r −0.956	Pearson r −0.956	Pearson r −0.956	Pearson r −0.956	Pearson r −0.956
Epitheloid GBM	*p* = 0.62	*p* = 0.06	*p* = 0.06	*p* = 0.06	*p* = 0.06	*p* = 0.06
	Pearson r 0.119	Spearman r −0.432	Spearman r −0.432	Spearman r −0.432	Spearman r −0.432	Spearman r −0.432
All GBM subtypes	*p* = 0.93	*p* = 0.93	*p* = 0.93	*p* = 0.93	*p* = 0.93	*p* = 0.93
	Spearman r −0.009	Spearman r −0.009	Spearman r −0.009	Spearman r −0.009	Spearman r −0.009	Spearman r −0.009

## Data Availability

The datasets generated and/or analyzed in this study are available upon request.

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
