# Peer review of "Prognostic Factors of Survival in Glioblastoma Multiforme Patients—A Retrospective Study"

_diagnostics, 2022, doi:10.3390/diagnostics12112630_

Round 1

Reviewer 1 Report

Thank you for the opportunity to review this manuscript. The analysis provided by the authors appears to be in agreement with prior similar studies. The manuscript is well written but there are issues in data analysis that need to be addressed. I would recommend that authors add correction for multiple comparison to their analysis. This will control for Type I errors, especially when the data set is small.

Also, if possible, I would recommend that authors add comparison based on involvement of dominant versus non-dominant hemisphere. Data regarding involvement of eloquent cortex if available would also add to the analyses. These factors limit the resection of the tumor and may possibly affect prognosis.

I would also recommend reformatting the abstract of this manuscript. The authors have presented a paragraph prior to the background that seems to summarize the abstract.

Author Response

Thank you very much for your thorough review! We deeply appreciate your effort to help us increase our paper's scientific value.

Please see below the modifications we made in order to comply with your requirements. We also submitted the new version of our review in the attachment.

1) “The manuscript is well written but there are issues in data analysis that need to be addressed. I would recommend that authors add correction for multiple comparison to their analysis. This will control for Type I errors, especially when the data set is small.”

Answer: The statistics program we used (Graph Pad) automatically applies post-hoc analysis in case a statistically significant result has been found when comparing means/medians among more than three samples (in terms of age, dimensions and so forth). However, we did not obtain any statistically significant difference. Nonetheless, we are extremely grateful for this observation as we did omit to mention these post-hoc tests when describing statistics part within the “Materials and methods” section. Please see the updated section between lines 107-111.

2) “I would recommend that authors add comparison based on involvement of dominant versus non-dominant hemisphere.”

Answer: Please see lines111-113,130-133, 151-154, 168-171, Table 2 and Figure 1c.

3) “I would also recommend reformatting the abstract of this manuscript.”

Answer: We erased the “simple summary”, as it is not mandatory in article format required by Diagnostics journal.

Reviewer 2 Report

The authors of "Prognostic factors of survival in glioblastoma multiforme patients – A retrospective study" have presented a study of GBM patients including clinical characteristics and tumor tissue analyses. They had the laudable goal of identifying prognostic factors that might aid in physicians' treatment strategies for this patient, but due to well explained limitations, the authors did not identify any novel prognostic factors.  The most important finding of the study seems to be that their patient population overall had a lower survival expectancy than the overall survival found with treatment using the Stupp regimen. Although this was explained by tumor size, it would be of interest to readers to compare this GBM size at time of diagnosis between this patient population and the populations from other published GBM studies. 

Minor comments: 1) The Authors include in the introduction, "Moreover, the intercellular transport of different species of circular RNA at molecular level supposedly induces radioresistance in GBM cells [2]." This finding from their research group, though interesting, is not a validated and established as are the other descriptions of GBM characteristics in the introduction, and should be removed. 2) The drug name temozolomide is the generic name for the drug and therefore it should not be capitalized in the manuscript. It is also misspelled at the top of page 2. 

Author Response

Thank you very much for your thorough review! We deeply appreciate your effort to help us increase our paper's scientific value.

Please see below the modifications we made in order to comply with your requirements. We also submitted the new version of our review in the attachment.

1) “The most important finding of the study seems to be that their patient population overall had a lower survival expectancy than the overall survival found with treatment using the Stupp regimen. Although this was explained by tumor size, it would be of interest to readers to compare this GBM size at time of diagnosis between this patient population and the populations from other published GBM studies.”

Answer: We cited additional studies in the “Discussions” section covering the influence of tumour size on overall survival. We also add the possible impact of involvement of dominant hemisphere on the poorer prognostic of our patients. Please see newly added lines 272-290.

2) “The Authors include in the introduction, "Moreover, the intercellular transport of different species of circular RNA at molecular level supposedly induces radioresistance in GBM cells [2]." This finding from their research group, though interesting, is not a validated and established as are the other descriptions of GBM characteristics in the introduction, and should be removed."

Answer: We erased that sentence. Please see the updated introduction (line 49).

3) “The drug name temozolomide is the generic name for the drug and therefore it should not be capitalized in the manuscript. It is also misspelled at the top of page 2.” 

Answer: Please see line 43 and 84.